# Self-Assembled Bimetallic Aluminum-Salen Catalyst for the Cyclic Carbonates Synthesis

**DOI:** 10.3390/molecules26134097

**Published:** 2021-07-05

**Authors:** Wooyong Seong, Hyungwoo Hahm, Seyong Kim, Jongwoo Park, Khalil A. Abboud, Sukwon Hong

**Affiliations:** 1Department of Chemistry, Gwangju Institute of Science and Technology, Gwangju 61005, Korea; optimistwy@gist.ac.kr (W.S.); hyungwoo@gist.ac.kr (H.H.); timmy369@gist.ac.kr (S.K.); 2Department of Chemistry, University of Florida, P.O. Box 117200, Gainesville, FL 32611-7200, USA; Jongwoo.Park@sk.com (J.P.); abboud@chem.ufl.edu (K.A.A.); 3SK Biotek, 325 Exporo, Yuseong-gu, Daejeon 34124, Korea; 4Gwangju Institute of Science and Technology, School of Materials Science and Engineering, Gwangju 61005, Korea

**Keywords:** aluminum, bimetallic catalyst, salen, urea, self-assembled, cyclic carbonate, epoxide, carbon dioxide

## Abstract

Bimetallic bis-urea functionalized salen-aluminum catalysts have been developed for cyclic carbonate synthesis from epoxides and CO_2_. The urea moiety provides a bimetallic scaffold through hydrogen bonding, which expedites the cyclic carbonate formation reaction under mild reaction conditions. The turnover frequency (TOF) of the bis-urea salen Al catalyst is three times higher than that of a μ-oxo-bridged catalyst, and 13 times higher than that of a monomeric salen aluminum catalyst. The bimetallic reaction pathway is suggested based on urea additive studies and kinetic studies. Additionally, the X-ray crystal structure of a bis-urea salen Ni complex supports the self-assembly of the bis-urea salen metal complex through hydrogen bonding.

## 1. Introduction

The conversion of carbon dioxide (CO_2_) into value-added chemicals (carbon capture and utilization, CCU) is an area that has been increasingly studied over recent decades [1]. The conversion reaction of CO_2_ and epoxides to cyclic organic carbonates has a 100% atom economy, and the cyclic carbonates are important intermediates for polymers [2,3], agrochemicals, pharmaceuticals [4], electrolytes in lithium-ion batteries [5], polar aprotic solvents [6,7], and temporal protecting groups for cis-diols [8]. Owing to the usefulness of cyclic carbonate compounds, various catalysts have been developed for the reaction between CO_2_ and epoxides [9,10,11,12,13,14,15].

Among the developed systems, bimetallic catalysis enhances catalytic performance in cyclic carbonate synthesis reactions. In pioneering studies, Lau and Tang independently reported single-framed hetero-bimetallic (Mn and Ru) [16] and homo-bimetallic (bi-Co) [17] catalysts for the coupling reaction between CO_2_ and epoxides. North group also reported a μ-oxo-bridged bimetallic salen aluminum catalyst [18,19]. Since then, various bimetallic catalysts have been developed, including single-framed bimetallic catalysts [16,17,20,21,22], bridged bimetallic catalysts [18,19,23,24,25,26], ligand-separated bimetallic catalysts [27,28], and covalent bond-tethered catalysts [29,30].

In some bimetallic catalytic reactions, two metal centers can cooperatively and simultaneously activate both reaction partners (i.e., epoxide and CO_2_), which leads to second-order reaction kinetics [31]. For example, North group reported bimetallic aluminum catalysts, where the μ-oxo-bridged salen aluminum catalyst showed a 10-fold increase in turnover frequency (TOF) compared to a monometallic salen aluminum catalyst (Figure 1a μ-oxo-bridged catalyst) [19]. In their proposed mechanistic scenarios, one aluminum center was coordinated by ammonium-supported CO_2_ and the other was coordinated by the epoxide [19,32]. North and co-workers also demonstrated that the μ-oxo-bridged salen aluminum catalyst remained active in the absence of an ammonium halide cocatalyst [33]. Later, North group developed the single framed bimetallic salen aluminum catalyst (Figure 1a) [34].

We previously reported self-assembling urea-functionalized salen cobalt catalysts for the hydrokinetic resolution of epoxides [35]. Those catalysts can self-assemble in solution through weak hydrogen bonding interactions [31,35]. Herein, we report that the self-assembling bimetallic strategy can also be applied to the salen-aluminum catalysts for the cyclic carbonate synthesis from CO_2_ and epoxides [36,37,38,39,40,41].

## 2. Results and Discussion

### 2.1. Catalyst Preparation

Bis-urea salen ligands were synthesized in five steps following a reported procedure. The ligands were prepared from commercially available 2-tert-butylphenol, and the overall yield was 44%. All intermediate compounds were checked by ^1^H NMR [35]. Metalation with aluminum chloride was performed by a previously reported procedure that involved a salen ligand and di-(tert-butyl) aluminum chloride (DIBAL-Cl) (Figure 2) [42]. The formation of the bis-urea salen aluminum catalyst was confirmed by high-resolution fast atom bombardment (FAB) mass spectrometry ([M − Cl]^+^, *m/z* = calc. 1027.3356, found 1027.3505) (See Appendix A for more details).

### 2.2. X-ray Crystallography

While we tried to grow a single crystal of a self-assembled bis-urea salen aluminum complexes, crystallization was unsuccessful. Instead, a single crystal of the bis-urea salen nickel complex was obtained by slow evaporation in *N,N*-dimethylformamide (DMF) (Figure 3, **4**). The Ni complex (**4**) has same ligand as bis-urea salen catalyst (3b). As a packed crystal, the Ni complex shows a parallel head-tail conformation. In this structure, intermolecular hydrogen bonding interactions between urea groups are observed (N–H•••O = 2.06, 2.08 Å) at both ends of the salen, and the two urea planes are significantly twisted (57.9(8)°). The metal–metal distance is measured as 5.3 Å.

The dimeric structure interacts with the neighboring dimer through extended urea-urea hydrogen bonds (Figure 3). However, in this association, only the alkyl urea N-H is involved in the formation of an intermolecular N–H•••O interaction of 2.05 Å with the neighboring dimer. The aryl urea N-H forms N–H•••O interactions of 2.28 Å with one DMF molecule. Furthermore, the urea–urea plane is twisted at an angle of 53.4(1)°. The metal–metal distance between neighboring dimers is measured as 4.9 Å (Figure 4). The crystal structure of the bis-urea salen nickel complex supports the bimetallic scaffold of the bis-urea salen aluminum catalyst.

### 2.3. Optimization of Cyclic Carbonates Synthesis Reaciton Conditions

The initial test reaction used for this study was the conversion of propylene oxide to propylene carbonate (Figure 5). The initial reaction was conducted in a closed system consisting of a Teflon sealed Schlenk flask at 1 bar of CO_2_ and 45 °C. The bis-urea salen aluminum catalyst (**3b**) exhibited a TOF that was 13 times higher than a TOF of a standard salen-aluminum catalyst (**3a**) with tetrabutylammonium bromide ((*n*-Bu)_4_N^+^Br^−^) (Table 1, Entries 1 and 2). Use of tetrabutylammonium iodide ((*n*-Bu)_4_N^+^I^−^) led to slightly higher TOF, compared to the use of TAB under the initial reaction conditions (1 bar of CO_2_ and 45 °C) (Table 1, Entries 2 and 3). To increase the TOF, the reaction was conducted in a 10 mL stainless steel bomb reactor. At 10 bar of CO_2_ and 90 °C, the bis-urea salen aluminum catalyst showed a higher TOF than previously reported monometallic salen aluminum catalyst (Table 1, Entries 4 and 5) and bridged salen bimetallic aluminum catalyst (Table 1, Entries 5 and 6). It is important to note that the urea moiety catalyst improves the TOF regardless of the conditions of the cyclic carbonate synthesis from CO_2_ and epoxide reaction system.

### 2.4. Cyclic Carbonate Synthesis Reactions for Various Epoxides

The epoxide substrate scope was studied for the conversion reactions of various terminal epoxides (**5a**–**5h**) to cyclic organic carbonates (**6a**–**6h**). The bis-urea salen aluminum catalyst was most active when alkyl- or hydroxyalkyl-substituted terminal epoxides were used (**5a**–**5c**, **5e**, and **5g**). The TOF was affected by the steric demands of the epoxides (**5a**–**5c**). Chloroalkyl-substituted epoxides (**5d**) and styrene oxide (**5f**) showed low TOFs. However, alkoxy-substituted epoxides (**5e**, **5g**, **5h**) showed higher activity than other terminal epoxides, as previously reported [27]. Use of low amounts of catalyst **3b** (0.02 mol%) and cocatalyst ((n-Bu)_4_N^+^I^−^, 0.02 mol%) were allowed under comparatively mild conditions (90 °C, pCO_2_ = 10 bar) for various epoxide substrates (Figure 6).

### 2.5. Origin of Beneficial Urea Effects

In the monomeric salen aluminum catalyst system, 0.08 mol% (4 equivalent with respect to the catalyst) of urea additive was added and tested under the optimized conditions; notably, the TOF did not increase. The free urea in the reaction system did not appear to affect the reaction (Figure 7) (Table 2, Entries 1 and 2).

We speculated that the TOF enhancement could occur owing to the bimetallic character enabled by the hydrogen bonding between urea groups of the ligand. Thus, we plotted a reaction rate vs. the changes in the amount of catalyst **3b**. The graph showed a clear second-order functional graph (R^2^ = 0.9977), suggesting a bimetallic reaction pathway (Figure 8) [35].

FTIR spectroscopy study was conducted to obtain more direct experimental evidence for self-association through urea-urea hydrogen bonding in solution. Bis-urea salen Al catalyst (**3b**) in THF was measured by FTIR at 25 °C. The FTIR experiments revealed the intensity of hydrogen bonded NH stretching vibration ($\tilde{\nu }$ = 3444 cm^−1^) increased with increasing concentration, and the intensity of free NH stretching vibration ($\tilde{\nu }$ = 3966 cm^−1^ and 3808 cm^−1^) decreased with increasing concentration. The results of FTIR experiments suggest intermolecular hydrogen bonding between bis-urea salen Al complexes in THF solution (Figure 9 See Appendix A for more details) [35].

## 3. Conclusions

In conclusion, the self-assembly strategy was successfully applied to the functionalized salen-aluminum catalysts for cyclic carbonate synthesis reaction from CO_2_ and epoxide. The bis-urea functionalized salen ligands were designed to self-assemble through urea-urea hydrogen bonding. The bimetallic urea salen aluminum complex showed an improved reaction rate (up to 13 times at 1 bar of CO_2_, 45 °C and up to 2.2 times at 10 bar of CO_2_, 90 °C) in the cyclic carbonate formation from epoxides with CO_2_. Free urea additive studies and kinetic studies were performed to confirm bimetallic reaction pathway. FTIR spectroscopy study provided an experimental evidence for urea-urea hydrogen bonding in solution. This work demonstrates that hydrogen bonding can be applied to the catalysts for cyclic carbonate synthesis reaction. It is important to note that the urea moiety improves the TOF regardless of the reaction conditions. Modifications of the ligand structures to further improve the catalyst are currently in progress.

## Figures and Tables

**Figure 1 molecules-26-04097-f001:**
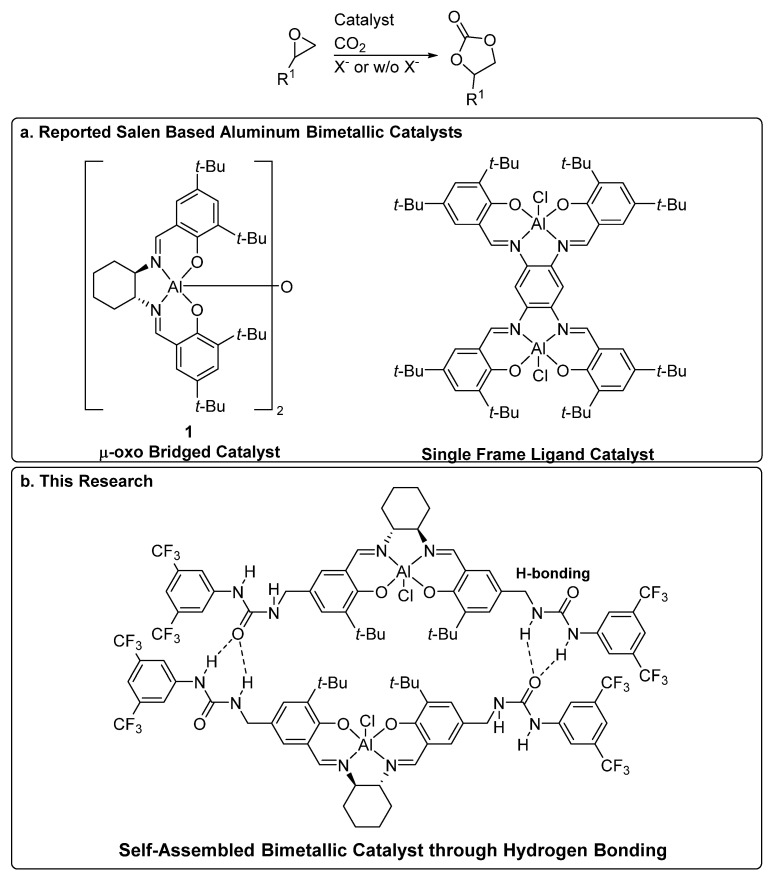
Bimetallic catalysts for cyclic carbonates synthesis. (**a**) Reported bimetallic salen catalysts. (**b**) This research: self-assembled bimetallic catalysts through hydrogen bonding.

**Figure 2 molecules-26-04097-f002:**
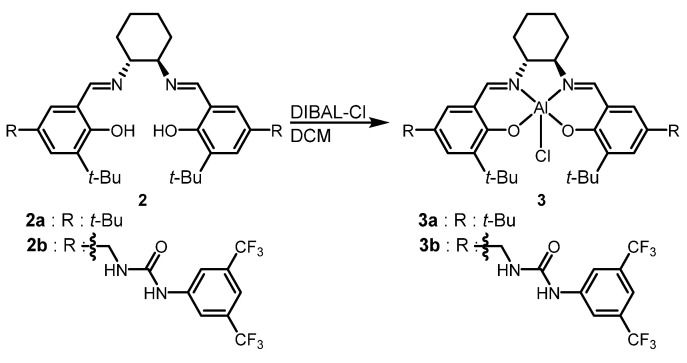
Synthesis of salen-aluminum catalysts.

**Figure 3 molecules-26-04097-f003:**
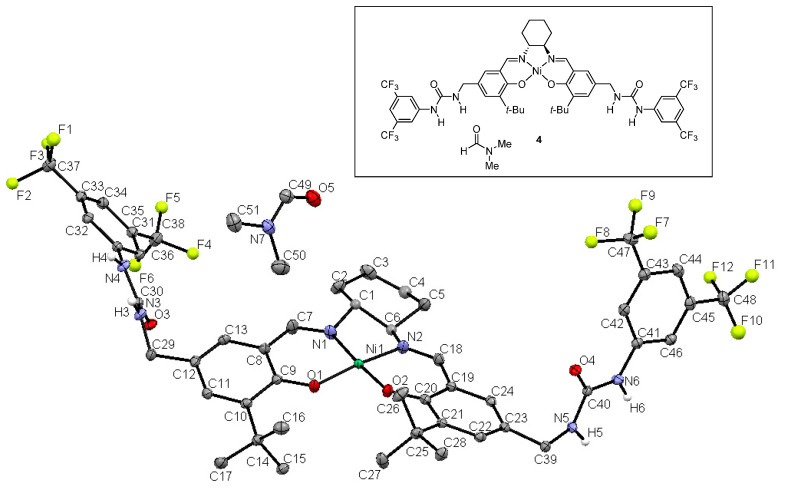
X-ray crystal structure of bis-urea salen nickel complex **4**.

**Figure 4 molecules-26-04097-f004:**
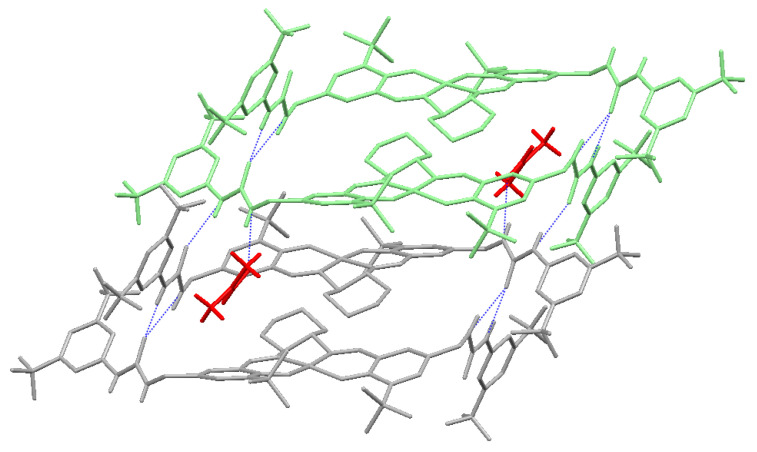
Hydrogen-bond packing structure of bis-urea salen nickel complex.

**Figure 5 molecules-26-04097-f005:**
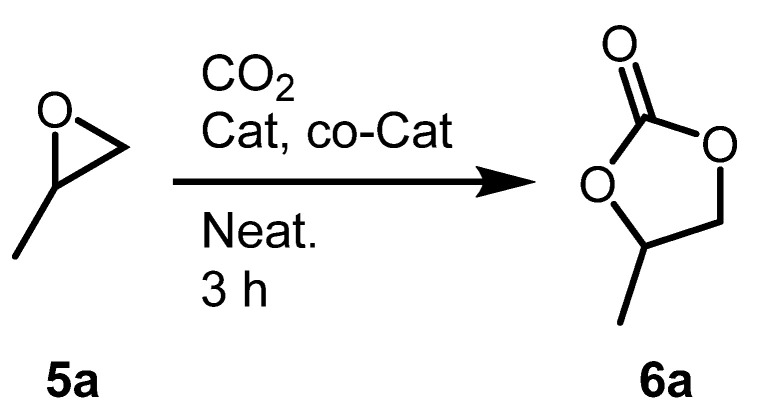
Cyclic carbonate synthesis using salen-Al catalysts and tetrabutylammonium halides.

**Figure 6 molecules-26-04097-f006:**
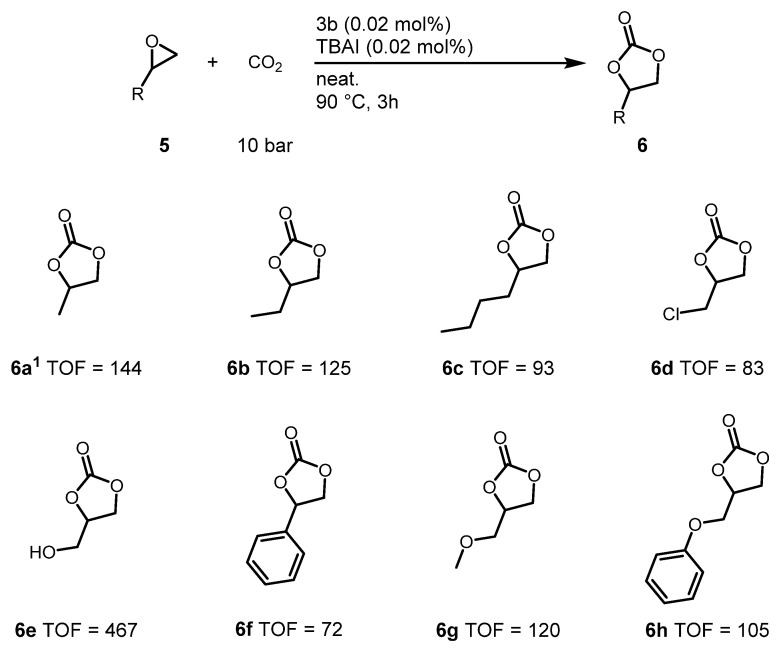
Scopes of epoxides. Reaction conditions: Epoxide (47 mmol), CO_2_ (10 bar), Catalyst (10 mg, 0.0094 mmol), (*n*-Bu)_4_N^+^I^−^ (3.5 mg, 0.0094 mmol), reaction was carried out in a 10 mL sealed stainless steel pressure bomb reactor. All TOFs are determined by ^1^H NMR using mesitylene internal standard (1.0 mmol, 0.1 mL). ^1^ 3.5 h reaction.

**Figure 7 molecules-26-04097-f007:**
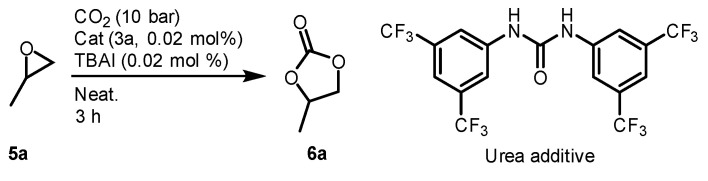
Urea additive study.

**Figure 8 molecules-26-04097-f008:**
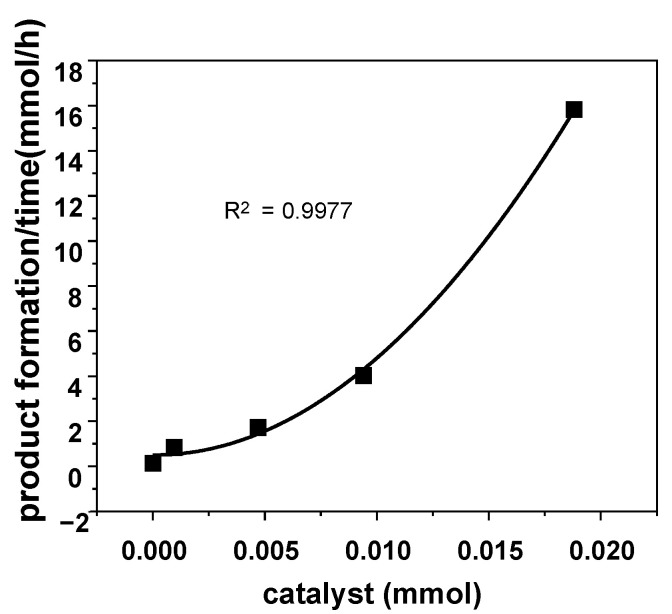
Kinetic analysis of the reaction order (Second order, R^2^ = 0.9977). Reaction conditions: Epoxide (47 mmol), CO_2_ (10 bar), (*n*-Bu)_4_N^+^I^−^ (3.5 mg, 0.0094 mmol), 90 °C, 3 h, reaction was carried out in a 10 mL sealed stainless steel pressure bomb reactor. All TOFs are determined by ^1^H NMR using mesitylene internal standard (1.0 mmol, 0.1 mL).

**Figure 9 molecules-26-04097-f009:**
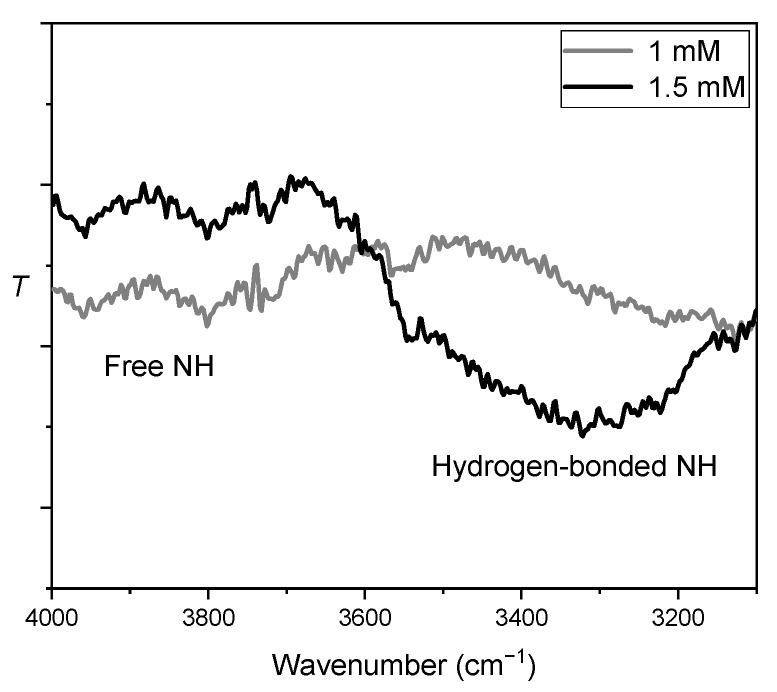
The NH stretching region of the FTIR spectra of **3b** in THF at two different concentration (1 mM (grey), 1.5 mM (black)) at 25 °C.

**Table 1 molecules-26-04097-t001:** Cyclic carbonate synthesis using salen-Al catalysts and tetrabutylammonium halides.

Entry	Catalyst (mol%)	Cocatalyst (mol%)	CO_2_ (Bar)	T (°C)	TOF (h^−1^) ^1^
1 ^2^	**3a** (1)	(*n*-Bu)_4_N^+^Br^−^ (1)	1	45	1
2	**3b** (0.02)	(*n*-Bu)_4_N^+^Br^−^ (0.02)	1	45	13
3	**3b** (0.02)	(*n*-Bu)_4_N^+^I^−^ (0.02)	1	45	17
4	**3a** (0.02)	(*n*-Bu)_4_N^+^I^−^ (0.02)	10	90	65
5	**3b** (0.02)	(*n*-Bu)_4_N^+^I^−^ (0.02)	10	90	144
6	**1** (0.02)	(*n*-Bu)_4_N^+^I^−^ (0.02)	10	90	47 ^3^

^1^ Determined by ^1^H NMR using mesitylene internal standard ^2^ 12 h reaction ^3^ TOF = TON/h·[Al].

**Table 2 molecules-26-04097-t002:** Urea additive study.

Entry	Urea Additive (mol%)	TOF (h^−1^)
1	-	64
2	0.08 mol%	65

## Data Availability

Not applicable.

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
