# Peer review of "Self-Assembled Bimetallic Aluminum-Salen Catalyst for the Cyclic Carbonates Synthesis"

_molecules, 2021, doi:10.3390/molecules26134097_

Round 1
Reviewer 1 Report
This experimental study describes the synthesis of a series of cyclic carbonates from carbon dioxide and different epicures catalyzed by bis-urea functionalized Al salen complexes. The idea, based on the crystallographic structure of a Ni analogous complex, is that the catalysts acts as a bimetallic species, due to intramolecular hydrogen binding which provides a supramoleular interesting structure. This hypothesis is also suffragated by the reaction order determination which is shown in Fig. 6 of the paper.
Although interesting, I think that the present study could fit better in a catalysis journal because .
in addition, the conclusions drawn by the authors are rather speculative and thus not fully convincing. Theoretical studies might be dcartied out to enforce the hypothesis.
Minor:
- no errors are indicated in Fig 6
- there are some typos... for example what is a shrink flask (supporting information)?
Author Response
Thank you for your helpful comments about our manuscript and interest of our research.
Comment 1 : in addition, the conclusions drawn by the authors are rather speculative and thus not fully convincing. Theoretical studies might be dcartied out to enforce the hypothesis
Response to comment 1 : We have redrafted the conclusion section to establish a clearer focus. (page 7, line 162-174 in revised manuscript)
Comment 2 : no errors are indicated in Fig 6
Response to comment 2 : It sould be better to have error bars in figure 6 for more accurate analysis, however, those type of kinetic analysis graphs have often been presented without error bars (see J. Am. Chem. Soc. 2021, 143, 7512, 7513, 7514. / J. Am. Chem. Soc. 2020, 142, 19231. / J. Am. Chem. Soc. 2001, 123, 1850, 1851. / Molecules 2021, 26, 431. / Angew. Chem. Int. Ed. 2021, 60, 775, 776. / Angew. Chem. Int. Ed. 2021, 60, 5270 et al.).
Comment 3 : there are some typos... for example what is a shrink flask (supporting information)?
Response to comment 3 : We checked our supporting information again, and rewrited the supporting information to correct typo. (Shrenk to Schlenk)
(page SI2-SI3 in revised SI)
Please check out our revision letter and marked manuscript too.
Thank you.

Reviewer 2 Report
This Seong, W.; Hahm, H.; Park, J.; Abboud, K.A.; Hong, S. Self-Assembled Bimetallic Aluminum-Salen Catalyst for the Cyclic Carbonates Synthesis. Paper describes the synthesis, characterization and catalytic behaviour of a Salen dialuminium catalyst in the presence of urea.
I do not recommended this manuscript for publication. The paper is based on the principal idea about the importance of non-covalent interactions in the catalytic behaviour of an Al salen compound. They proposed that this non-covalent interaction in the ligand is caused by H intermolecular bonding due to the urea presence in the reaction media. I think to affirm this it could be necessary doing more studies in solution such us NMR Diffussion studies or a possible exchange between salen ligand and the urea proligand that affects the structure of the initial catalyst. I do not found an NMR study of the behaviour for the Al compound as the same time of the urea in solution.
Moreover, the authors reclaim the proposed association of the ligand due to the formation of an intermolecular N–H•••O interaction described for an analogous Ni complex. I have various questions at this point: Ni and Al are different in their chemistry, so in solid state, although with the same ligand, we can expect a different behaviour too. Moreover, the interaction is observed in solid state but in solution it could be possible that did not exist. The author has a similar article (DOI:10.1002/anie.201107785) based on a similar idea but for transition metal compounds.
Another questions are:
- It is not clear which catalyst is 3a (standard salen- aluminum catalyst (3a) is written) or 3b. The structures or the formula could be of interest.
- Concerning to typing errors:
Line 92: reaction must be written
Line 99: TAB is written instead of TBAB for tetrabutyl ammonium bromide
Author Response
Thank you for giving us the opportunity to strengthen our manuscript with your valuable comments.
Comment 1 : I think to affirm this it could be necessary doing more studies in solution such us NMR Diffussion studies or a possible exchange between salen ligand and the urea proligand that affects the structure of the initial catalyst. I do not found an NMR study of the behavior for the Al compound as the same time of the urea in solution.
Response to comment 1 : To accommodate this comment, we have included a new FT-IR spectrum figure (Fig 7.) to illustrate that hydrogen bonding occurred in the solution. (page 6,line 151-158 in revised manuscript), (page 7, Figure 7. And line 159-161 in revised manuscript), (page SI-2, Line 13-17 in revised SI)
Comment 2 : Moreover, the authors reclaim the proposed association of the ligand due to the formation of an intermolecular N–H•••O interaction described for an analogous Ni complex. I have various questions at this point: Ni and Al are different in their chemistry, so in solid state, although with the same ligand, we can expect a different behaviour too. Moreover, the interaction is observed in solid state but in solution it could be possible that did not exist. The author has a similar article (DOI:10.1002/anie.201107785) based on a similar idea but for transition metal compounds.
Response to comment 2 : We agree that Ni and Al are different in their chemistry even in solid state so we tried to grow a single crystal of a bis-urea salen aluminum complexes but the crystallization was unsuccessful. But we provided FTIR spectroscopy data of solid bis-urea salen aluminum complex in SI and the hydrogen bonded NH stretching vibration peak was shown. We believe that the FTIR spectrum of the bis urea salen aluminum complex powder could be an indirect evidence for hydrogen bonding of our catalyst. Also we conducted additive study in our manuscript (page 6. Table 2. In revised manuscript). Unlike previous results in previous article (DOI:10.1002/anie.201107785), the TOF did not increase. So, the urea does not seem to be involved in activation of the reagent (bifunctional monomeric mechanism), which is different from the current result.
(page SI-44,45 in revised SI)
Comment 3 : It is not clear which catalyst is 3a (standard salen- aluminum catalyst (3a) is written) or 3b. The structures or the formula could be of interest.
Response to comment 3 : We redraw fig 2. To clarified which catalyst is 3a or 3b. (page 3, Figure 2. in revised manuscript)
Comment 4 : Concerning to typing errors: Line 92: reaction must be written, Line 99: TAB is written instead of TBAB for tetrabutyl ammonium bromide
Response to comment 4 : We rewrite line 92 (now it is the line 96 in revised manuscript).(page 4, line 96 in revised manuscript)And use more clear term (n-Bu)4N+Br- and (n-Bu)4N+I- instead of TBAB or TBAI to avoid unclarity. (page 4, line 101 and 102 in revised manuscript) (page 4, line 111-120 in revised manuscript) (page 5, Table 1. And Figure 5. in revised manuscript) (page 6, Figure 6. line 148 in revised manuscript)
please check out attached file which is our revision letter. Thank you.

Reviewer 3 Report
The manuscripts describes synthesis, characterisation and performance of bimetallic Aluminum-Salen catalyst for cyclic carbonates synthesis from epoxides and CO2. The study reports TOF of their catalyst as three times higher than that of a μ-oxo-bridged catalyst, and 13 times higher than that of a monomeric salen aluminum catalyst.
They motivate their study CO2 utilization for cyclic carbonates which are important in polymer synthesis and other important applications like drugs, lithium-ion batteries to name but a few.
In introduction part there is a short review of already known catalysts of cyclic carbonate synthesis reactions.
As I understand the manuscript is extension of previous study concerning salen cobalt catalysts to the new catalyst i.e. salen-aluminum compound.
The structure was analysed with X-ray crystallography. However the monocrystal of bis-urea salen aluminium complex was not possible to obtain. Instead analogous structure i.e. complex of Ni was used. The activity of catalyst was measured at temperatures 45 and 90C and pressure of 1 and 10 bars.
Authors expect further improvement of catalytical activity by by modification the structure of ligands.
In my opinion the manuscript is worth of publication. It is very short but it is neatly written (I haven’t found any typo) and an extended information are provided in supplementary materials.
Author Response
Thank you for your comment that “the manuscript is worth of publication. It is very short but it is neatly written”.
We have redrafted the conclusion section to establish a clearer focus.
(page 7, line 162-174 in revised manuscript)
please check out the attached file which is the revision letter.
Thank you.

Round 2
Reviewer 1 Report
The authors have addressed my points. Some of them indeed would require significant additional work, because the conclusions still remain a bit speculative in my opinion. Anyway there is no concern about the included experiements and results.